



# Learning Extreme Vegetation Response to Climate Forcing: A Comparison of Recurrent Neural Network Architectures

Francesco Martinuzzi[1,2,3], Miguel D. Mahecha[1,2,3,4], Gustau Camps-Valls[5], David Montero[2,3], Tristan Williams[5], and Karin Mora[2,3]

[1]Center for Scalable Data Analytics and Artificial Intelligence, Leipzig University, Leipzig, Germany
[2]Institute for Earth System Science & Remote Sensing, Leipzig University, 04103 Leipzig, Germany
[3]Remote Sensing Centre for Earth System Research, Leipzig University, Leipzig, Germany
[4]German Centre for Integrative Biodiversity Research (iDiv), Leipzig, Germany
[5]Image Processing Laboratory (IPL), Universitat de València, València, Spain

**Correspondence:** Francesco Martinuzzi `martinuzzi@informatik.uni-leipzig.de`

**Abstract.** Vegetation state variables are key indicators of land-atmosphere interactions characterized by long-term trends, seasonal fluctuations, and responses to weather anomalies. This study investigates the potential of neural networks in capturing vegetation state responses, including extreme behavior driven by atmospheric conditions. While machine learning methods, particularly neural networks, have significantly advanced in modeling nonlinear dynamics, it has become standard practice

5   to approach the problem using recurrent architectures capable of capturing nonlinear effects and accommodating both long and short-term memory. We compare four recurrence-based learning models, which differ in their training and architecture: 1) recurrent neural networks (RNNs), 2) long short-term memory-based networks (LSTMs), 3) gated recurrent unit-based networks (GRUs), and 4) echo state networks (ESNs). While our results show minimal quantitative differences in their performances, ESNs exhibit slightly superior results across various metrics. Overall, we show that recurrent network architectures

10   prove generally suitable for vegetation state prediction yet exhibit limitations under extreme conditions. This study highlights the potential of recurrent network architectures for vegetation state prediction, emphasizing the need for further research to address limitations in modeling extreme conditions within ecosystem dynamics.



## 1  Introduction

The recent increase in atmospheric $CO_2$ concentrations only partly reflects anthropogenic emissions, as oceans and land ecosys-
tems contribute to the carbon uptake (Eggleton, 2012; Le Quéré et al., 2018; Canadell et al., 2021). Forests and other terrestrial
ecosystems absorb nearly a third of human-made emissions and establish an essential negative feedback within the global
carbon cycle (Friedlingstein et al., 2006; Le Quéré et al., 2009). However, during extreme events such as persistent droughts
and heatwaves, ecosystems may release more $CO_2$ into the atmosphere than they absorb due to suppressed photosynthesis
(von Buttlar et al., 2018; Sippel et al., 2018). Variations to the frequency and intensity of these events can lead to long-lasting
environmental modifications, contributing to positive feedback loops that aggravate climate warming (Reichstein et al., 2013).
For example, increases in drought intensities have been consistently linked to excess tree mortality (Grant, 1984; Fensham and
Holman, 1999; Liang et al., 2003; Dobbertin et al., 2007), negatively impacting carbon sequestration (Van Mantgem et al.,
2009). The frequency, intensity, and duration of extremes over the next few decades are expected to increase compared to pre-
vious decades (Seneviratne et al., 2021). Therefore, understanding how vegetation responds to climate drivers becomes crucial
in land-atmosphere modeling (Mahecha et al., 2022).

The vegetation response changes over time, showing seasonal patterns and long-term trends (Slayback et al., 2003; Mahecha
et al., 2010; De Jong et al., 2011, 2012; Linscheid et al., 2020). This variability is influenced by climate variables such as
radiation, temperature, and precipitation, which affect vital biosphere processes such as photosynthesis. These meteorolog-
ical variables create a range of conditions for the vegetation, from optimal to stressful (Nemani et al., 2003; Seddon et al.,
2016). However, the relationship between climate and biosphere involves complex interactions due to the nonlinear response
of vegetation to climate drivers (Foley et al., 1998; Zeng et al., 2002; Papagiannopoulou et al., 2017). Furthermore, ecosystems
exhibit memory effects (Johnstone et al., 2016; Pappas et al., 2017) that can put their long-term resilience at risk (De Keers-
maecker et al., 2016). For instance, extreme heatwaves can negatively impact leaf growth and development that, when coupled
with drought conditions, can lead to tree mortality (Teskey et al., 2015). Extreme perturbations can cause irreversible damage
(Scheffer et al., 2001) and reduce an ecosystem's resilience (Ghazoul et al., 2015). These factors collectively contribute to the
challenge of predicting the vegetation and climate system.

Traditionally, terrestrial biosphere models have played a pivotal role in simulating the impact of climate variability, for
example, in land carbon fluxes (Sitch et al., 2003; Krinner et al., 2005). Process-based models are inherently complex and
demand substantial computational resources (Watson-Parris, 2021). Despite their robust foundation in physical laws, they
sometimes fall short in mirroring the complex dynamics of land-atmosphere interactions accurately (Papale and Valentini,
2003). In response, there has been a growing reliance on machine learning (ML) techniques in Earth sciences (Zhu et al.,
2017; Tuia et al., 2023). These methods represent powerful modeling tools, able to find patterns in data that process-based
models may not be able to capture. As a consequence, applications of ML models in land-atmosphere interactions are wide-
ranging, from local-to-global flux upscaling (Papale et al., 2015; Jung et al., 2020) to the prediction of ecosystem states (Kang
et al., 2016; Zhang et al., 2021; Peng et al., 2022). More specifically, recurrent neural networks (RNNs) represent suitable
architectures for modeling complex Earth system dynamics (Reichstein et al., 2019; Camps-Valls et al., 2021b) due to their



ability to encode nonlinear temporal dependencies (Bengio et al., 1994; LeCun et al., 2015) and capacity to retain information from past inputs (Elman, 1990).

However, RNNs have technical challenges associated with gradient-based training, including the issues of vanishing and
exploding gradients, which impede network convergence (Hochreiter, 1998; Pascanu et al., 2013). To tackle these problems, specialized RNN architectures have been developed. Long short-term memory (LSTM) networks (Hochreiter and Schmidhuber, 1997) maintain the gradient-based training of the original RNNs while addressing these problems through gating mechanisms. Another architecture, the gated recurrent unit (GRU) (Cho et al., 2014), further refines the LSTM approach, providing comparable results with computational efficiency (Chung et al., 2014). In contrast, echo state networks (ESNs) employ a dis-
tinct approach by training only the last layer through linear regression (Jaeger, 2001). The absence of derivatives guarantees non-vanishing or exploding gradients, offering an alternative training solution to gating. The improvements provided by both the gated models and the ESNs allowed the application of these models to different tasks such as rainfall-runoff modeling (Kratzert et al., 2018; Gauch et al., 2021), sea surface temperature estimation (Zhang et al., 2017; Walleshauser and Bollt, 2022) and chaotic systems forecasting (Pathak et al., 2017, 2018; Vlachas et al., 2020; Chattopadhyay et al., 2020; Gauthier
et al., 2022) among others.

Expanding on the utility of RNNs, particularly LSTM networks, recent studies have demonstrated their effectiveness in addressing specific challenges within land-atmosphere interactions, modeling land fluxes from meteorological drivers (Reichstein et al., 2018). Further studies reinforced the suitability of RNN approaches for land-atmosphere interactions (Chen et al., 2021). In Besnard et al. (2019), the authors employ LSTMs to predict land fluxes from remote sensing data and climate variables
to explore the memory effects of vegetation. Additionally, the work provides a comparison with random forests, showing the better performance of deep learning models in this task. Further explorations in dynamic memory effects with LSTMs have been carried out by Kraft et al. (2019). Given the recent findings on the utility of RNNs and LSTMs in studying dynamics, it is timely to investigate if these tools can accurately predict extreme biosphere dynamics.

Can recurrent neural networks learn the vegetation's extreme responses to climate drivers? Recurrent architectures can
embed the dynamics of target systems into their higher dimensional representation (Funahashi and Nakamura, 1993; Hart et al., 2020). When considering the nature of extremes in dynamical systems, they can be understood as specific regions of the phase space (Farazmand and Sapsis, 2019). Combining this with the embedding abilities of RNNs, offers an explanation for the observed efficacy of ESNs and LSTMs in learning extreme events within controlled environments, or "toy models" (Srinivasan et al., 2019; Lellep et al., 2020; Pyragas and Pyragas, 2020; Ray et al., 2021; Meiyazhagan et al., 2021; Pammi et al., 2023).
However, the applicability of these findings to land-atmosphere interactions remains unclear. Unlike the systems investigated in previous studies, biosphere dynamics are characterized by stochasticity (Dijkstra, 2013) while their measurements present noise (Merchant et al., 2017). Therefore, exploring how recurrent architectures perform in the specific context of vegetation extremes in ecosystem dynamics is imperative.

To address this question, we investigate the ability of various recurrent architectures to model the response of vegetation
states, i.e., spectral reflectance indices, to climate drivers. Vegetation interacts with sunlight, showing specific spectral responses that can be altered during extreme events such as heat waves. Vegetation indices, obtained from the spectral response through





linear or nonlinear transformations, are used to quantify these changes, isolating vegetation properties from other influences such as soil background (Zeng et al., 2022); detailed lists of these indices are provided in Zeng et al. (2022) and Montero et al. (2023). Our study focuses on the normalized difference vegetation index (NDVI) (Rouse et al., 1974), which indicates vegetation greenness. To build a model that can accurately predict NDVI responses to climate conditions, we use climate-related variables, such as temperature and precipitation, as inputs.

The goal of this study is threefold: 1) to further solidify the viability of the recurrent neural network approach to model ecosystem dynamics, 2) to investigate whether these models can capture extreme vegetation responses to climate forcing, and finally, 3) to investigate whether a specific RNN architecture is more suited for these tasks. To evaluate the performance of the models, we use metrics such as the normalized root mean squared error (NRMSE) and symmetric mean absolute percentage error (SMAPE) in conjunction with information theory-based measures, namely the entropy-complexity (EC) plots (Rosso et al., 2007). The latter approach quantifies each model's ability to capture the target dynamics. Using the model's residuals, defined as the difference between the prediction and the actual signal, the EC plots return an intuition of the model's ability to capture dynamics beyond the seasonality (Sippel et al., 2016). Additionally, we investigate whether recurrent architectures can effectively capture vegetation responses to extreme events. To our knowledge, no study has compared the performance of recurrent models in the context of extreme events or ecosystem dynamics.

The remainder of this paper is structured as follows. In Sect. 2.1, we present the data we used, including the site selection process and pre-processing steps. Next, in Sect. 2.2, we describe the architecture of the recurrent neural networks and formalize the task. Following this, Sect. 2.3 and Sect. 2.4 give a background on the methods of backpropagation training and echo state networks, respectively. In Sect. 2.5 we describe the procedure to identify extreme events in the NDVI time series. We detail the metrics used in the study in Sect. 2.6. More specifically, in 2.6.1, we introduce NRMSE and SMAPE; in 2.6.2, we illustrate the EC plots; finally, in 2.6.3, we describe the metrics used for evaluating model performance on predicting extreme events. We show our results in Sect. 3. In Sect. 3.1 we compare model performances using NRMSE and SMAPE. Additionally, in Sect. 3.2 we show the results for the EC plots. Finally, we illustrate the models' capability to predict extreme events in 3.3. We draw conclusions and discuss broader implications in section 4.

## 2 Methods

### 2.1 Data and Pre-processing

We used optical remote sensing data of forest sites to measure biosphere dynamics, specifically the normalized difference vegetation index (NDVI) (Rouse et al., 1974), which we define as the "target" variable. However, employing NDVI presents drawbacks (Camps-Valls et al., 2021a). Namely, it has a saturating and nonlinear connection to green biomass and responds to greenness rather than the actual plant photosynthesis process. Despite these limitations, this index has been used successfully for various purposes, including, but not limited to, evaluating ecosystem resilience (Yengoh et al., 2015) and tracking the decline of vegetation greenness in the Amazon forests (Hilker et al., 2014). Additionally, NDVI was shown to be a good indicator of vegetation response to extreme climate events (Liu et al., 2013).



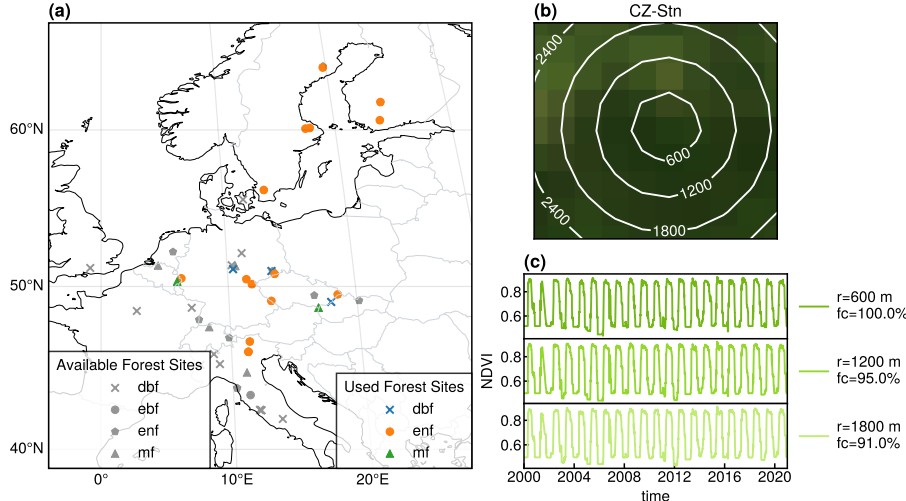

**Figure 1. Chosen locations and pre-processing** In **(a)**, we show the available and used locations for the study and their forest type. We used locations with more than 80% forest cover, not using the remaining ones depicted in gray in this figure. In **(b)**, we show the process of including additional pixels into the cube for an example location (CZ-Stn). While different radii are shown, we use all the pixels in a single cube. Finally, in **(c)**, we show the NDVI signal corresponding to the mean of the pixels in the area with radius $r$ previously shown for the same example location. Additionally, it also indicates the percentage of the pixels that are flagged as forest (fc).

We used NDVI values obtained from the moderate resolution imaging spectroradiometer (MODIS) in the FluxnetEO dataset v1.0 (Walther et al., 2022). This dataset is a multi-dimensional array of values referred to as a data cube (Mahecha et al., 2020). It comprises a collection of labeled univariate time series at the geographic location of eddy covariance (EC) towers. EC towers are specialized measurement stations designed to capture and quantify land-atmosphere fluxes and meteorological conditions, providing insights into ecosystem health and functionality (Aubinet et al., 2012). The FluxnetEO datasets are gap-filled using

measurements from EC towers, thus providing a higher resolution product in time and space compared to the raw MODIS data.

The data cubes in the dataset present a spatial resolution of 500 m and a daily temporal resolution, covering the period of 2000-2020 (inclusive). The cubes span a pixel of size $3\,\text{km} \times 3\,\text{km}$ centered on the EC towers in each location. We further pre-processed the data by averaging the NDVI for each timestep as the mean of all the pixels within the cube, see 1. Additionally, we smoothed the signal using a Savitzky-Golay filter (Savitzky and Golay, 1964; Steinier et al., 1972) with a 7-day window

and a polynomial order of 3 (Chen et al., 2004) to eliminate any potential artifacts caused by noise.

We selected study sites based on their forest cover percentage, ensuring over 80% forest cover in each cube, see 1. The forest masks were obtained from the Copernicus Global Land Service (CGLS) product (Buchhorn et al., 2020), which has a resolution of 100 m. These sites represent three different forest types: evergreen broadleaved forests (EBF), mixed forests (MF), and deciduous broadleaved forests (DBF). We chose to study forest sites for their importance to the carbon cycle and

because they are the least affected by human influence at a daily time scale. Consequently, we employed the described approach,





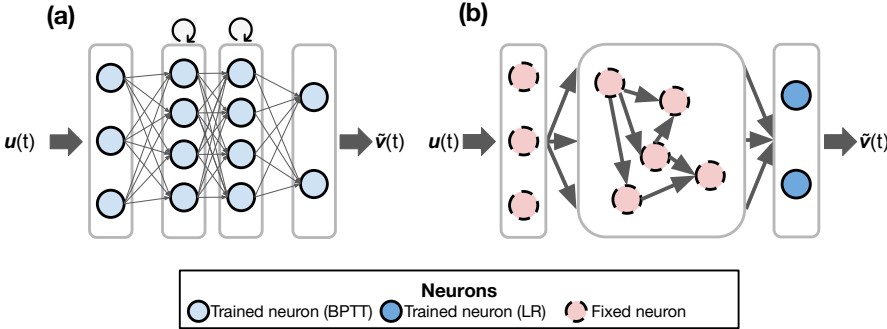

**Figure 2. Training methods.** Both diagrams illustrate input and predicted data denoted by $\boldsymbol{u}(t)$ and $\tilde{\boldsymbol{v}}(t)$, respectively. Diagram **(a)** presents the training methodology of ESNs. The initial two layers are randomly generated and remain untrained, while only the final layer undergoes one-shot training via linear regression (LR). In contrast, diagram **(b)** portrays the conventional approach used by RNNs, GRUs, and LSTMs, involving backpropagation through time (BPTT). Training encompasses the input layer (comprising three neurons in this instance), stacked recurrent layers (two layers, each with four neurons), and the output layer (two neurons). All these neurons are trained through backpropagation. The circular arrow atop signifies the recurrent layers. The number of neurons and internal recurrent layers serves visualization purposes only.

detailed in Fig. 1, to minimize further imperfections in the vegetation signal caused by human intervention. As a result of this selection criterion, the number of study sites is reduced from 42 to 20.

In this study, we used climate variables as input variables to predict the target variable. Following machine learning terminology, we will refer to them as "features." We selected air temperature (mean, minimum, and maximum), mean sea level pressure, mean global radiation, and precipitation as features. The climate data was obtained from the E-OBS product v26.0e (Cornes et al., 2018). Based on in situ observations, this dataset is spatially interpolated to cover most of the European continent. The spatial resolution is 0.1 degrees (11.1 km × 11.1 km), and the temporal resolution is daily. The time length of the feature variables is identical to the target variable and spans from 2000 to 2020 (inclusive).

## 2.2 Approach and Models

We aim to learn the NDVI behavior of forests as a proxy for ecosystem response to climate drivers. We use temperature (mean, minimum, maximum), precipitation, pressure, and radiation. We assume that the knowledge of the target variable is constrained to a certain period, after which only the features are available. Our goal is to build a model that, using those features, can predict the target variable. This is obtained by training the models on the available time interval, which comprises the years 2000-2013 (inclusive) for this study. After the training, the models only use feature variables to predict the NDVI for the remaining period, 2014-2020. Further details for the training setup can be found in Appendix B.

The task can be formalized as follows. We aim to approximate the target variable $\boldsymbol{v}(t) \in \mathbb{R}^{d_v}$ using input data $u(t)$. In the context of this study, we retain a unidimensional approach with $d_v = 1$, given that we have a single target variable, the NDVI.





We assume that both $\boldsymbol{u}(t)$ and $\boldsymbol{v}(t)$ are components of the same dynamical system. Under this assumption, the behavior $\boldsymbol{v}(t)$ is influenced by $\boldsymbol{u}(t)$, allowing us to leverage the information from $\boldsymbol{u}(t)$ to estimate $\boldsymbol{v}(t)$. Our setup consists of two sets of data,

$\boldsymbol{u}(t)$ and $\boldsymbol{v}(t)$, which can be measured over a given time period $t \in \{1, 2, \ldots, T\}$. After time $T$ only $\boldsymbol{u}(t)$ remains measurable, and the goal of the recurrent architectures is to create a model efficient in modeling $\boldsymbol{v}(t)$ based solely on the available $\boldsymbol{u}(t)$ data (Lu et al., 2017). In control theory, a model that can predict $\boldsymbol{v}(t)$ based on $\boldsymbol{u}(t)$ data is called "observer" (Hermann and Krener, 1977; Ogata et al., 2010). Despite having different training methods, all the models in this study are built to perform this task.

We consider four different recurrent architectures: recurrent neural networks (RNNs), long short-term memory based net-

works (LSTMs), gated recurrent unit-based networks (GRUs), and echo state networks (ESNs). Each of these models presents an internal state $\boldsymbol{x}(t) \in \mathbb{R}^{d_x}$, which encodes the temporal dependencies of the input data $\boldsymbol{u}(t) \in \mathbb{R}^{d_u}$, where $d_u = 6$ in our study, representing the dimension of the input data. The internal size $d_x$ is chosen to be $d_x > d_u$. A defining feature of these architectures is the recursive transmission of internal states, facilitating historical data retention as the model progresses through subsequent steps. This evolution of the generic RNN model is given by (Sutskever, 2013)

$$\boldsymbol{x}(t) = H(\boldsymbol{x}(t-1), \boldsymbol{u}(t); \theta), \tag{1}$$

where $\theta$ represents the weights and biases of the model, also called parameters, and $H$ represents a generic RNN update function.

### 2.3 Training - Backpropagation Through Time

Deep learning models like feed-forward neural networks (FFNN) adjust their weights at each training step $t \in \{1, 2, \ldots, T\}$

using a method called backpropagation (BP) (Rumelhart et al., 1986). BP relies on a "loss function" $\mathcal{L}$ given by $\mathcal{L}(\theta) = \sum_{t=1}^{T} \mathcal{L}_t(\theta)$, where $\theta$ stands for the network's parameters. Leveraging the loss function, BP minimizes the difference between the model's predicted and actual output by adjusting the model's weights. To do this, BP calculates the gradient of the loss function with respect to each weight and then updates the network weights in a direction that minimizes the loss. One of the most common approaches to minimizing the loss function is stochastic gradient descent (SGD) (Bottou, 2012). However, one

of BP's limitations is that it does not account for time dependencies.

For time-dependent models, such as RNNs, LSTMs, and GRUs, handling sequential data poses additional challenges. Backpropagation through time (BPTT) (Rumelhart et al., 1986) is a specialized training method designed for these architectures (Werbos, 1990). Central to BPTT is the notion of "unrolling" the network over time, effectively transforming it into a FFNN where backpropagation can be applied. This allows the model to calculate errors and update weights across the entire sequence,

making it possible to capture long-term dependencies and patterns in time series data.

However, applying BPTT across the entire sequence can be computationally intense and time-consuming. Moreover, it often reduces the error to a very small amount by the end of the sequence. To avoid this issue, we use a truncated version of BPTT, limiting the backpropagation to a fixed number of steps, denoted by $k$ (Williams and Zipser, 1995), which is smaller than the total number of training time steps, $T$ (Aicher et al., 2020).



This truncated approach ensures efficiency but requires transferring the last hidden state from the truncated section to the initial state in the following sequence. This step maintains some memory and ensures the network's training process continuity.

The final output of all the models considered in this study comes from a feed-forward layer. The parameters of this layer are also trained using BP. The following equation describes the feed-forward layer:

$$\tilde{\boldsymbol{v}}(t) = \sigma(\mathbf{W}^v \boldsymbol{x}(t) + \boldsymbol{b}^v), \tag{2}$$

where $\tilde{\boldsymbol{v}}(t) \in \mathbb{R}^{d_v}$ is the predicted output, $\boldsymbol{x}(t) \in \mathbb{R}^{d_x}$ is the hidden state of the model at time $t$, $\mathbf{W}^v \in \mathbb{R}^{d_v \times d_x}$ is the weight matrix and $\boldsymbol{b}^v \in \mathbb{R}^{d_x}$ is a bias vector. Additionally, $\sigma$ represents the activation function. This procedure is illustrated graphically in 2a. Finally, the full details of the models are given in appendix A1 for the RNNs, A2 for the LSTMs, and A3 for the GRUs.

The RNNs, LSTMs, and GRUs have been implemented using the `PyTorch` library (Paszke et al., 2019) accessed through `Skorch` (Tietz et al., 2017).

## 2.4    Training - Echo State Approach

Echo state networks (ESNs, Jaeger, 2001), along with liquid state machines (Maass et al., 2002), belong to the larger family of reservoir computing (RC) models, based on a shared theoretical background (Verstraeten et al., 2007). The fundamental idea of ESNs is to project the training data into a higher-dimensional, nonlinear system named the "reservoir" through an input layer. This process transforms the input data into vectors called "states." After the data passes through the reservoir, the states
are collected. The model is then trained by regressing these states against the target data.

More specifically, the ESNs have three layers: an input layer $\mathbf{W}_{\text{in}}$, a reservoir layer $\mathbf{W}$ and an output layer $\mathbf{W}_{\text{out}}$. A distinctive feature of ESNs is that the weights in the input and reservoir layers are fixed. These weights are generated randomly and do not change during training. This approach contrasts with conventional neural networks, where weights are continuously updated during training to reduce errors. For each training time step $t \in \{1, 2, \dots, T\}$, the hidden states, indicated as $\boldsymbol{x}(t)$, are preserved
and accumulated in a matrix $\mathbf{X} \in \mathbb{R}^{d_x \times T}$. Indicated as a "state matrix," this matrix effectively represents the system's dynamics. The last layer of the ESN is obtained through a linear regression operation that uses the states matrix to generate a feed-forward layer, creating the network's output layer:

$$\mathbf{W}_{\text{out}} = \mathbf{Y}^{\text{target}} \mathbf{X}^\top (\mathbf{X}\mathbf{X}^\top + \beta\mathbf{I})^{-1} \tag{3}$$

where $\mathbf{I}$ is the identity matrix and $\beta$ is a regularization coefficient. The matrix $\mathbf{Y}^{\text{target}} \in \mathbb{R}^{d_v \times T}$ is generated with the desired
output $\boldsymbol{v} \in \mathbb{R}^d_v$ stacked column-wise. While layers of recurrent models trained through BPTT can be stacked, ESNs are usually computed from a single inner layer (reservoir).

The construction and training of the ESN allow for the faster computational time of all the proposed models while also solving the vanishing and exploding gradients since no derivatives are taken at any step of the process. An illustration of the ESN approach to training is provided in Fig. 2b. The full details for the ESNs are given in appendix A4. For the implementations
of the ESNs, we relayed on the Julia package `ReservoirComputing.jl` (Martinuzzi et al., 2022).



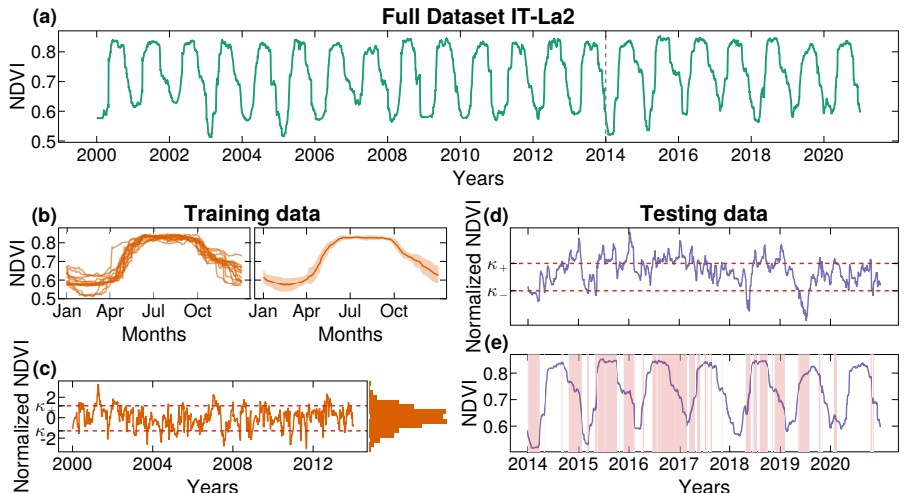

**Figure 3. Definition of extreme events.** In **(a)**, the NDVI of the dataset is plotted at an example location, IT-La2. The gray dotted line in 2014 represents the separation between training and testing data. In **(b)**, we show each yearly cycle of the training data: on the left are the full signals, and on the right are the mean and standard deviation (SD). In **(c)**, we show the result of the normalization of the training dataset using the mean and SD obtained in the previous step. At this stage, we also define the quantile (fixed at 0.90 in this example) and define the values of the thresholds $\kappa_+$ and $\kappa_-$. These values are then used in **(d)**, where we identify the extremes in the normalized testing data. The normalization is also done using the mean and SD of the training data. Finally, in **(e)**, the testing data is again shown in its raw form to showcase the extreme response of the vegetation.

## 2.5 Anomalies and Extremes

In this study, we adopt the approach outlined by Lotsch et al. (2005) to define anomalies based on the seasonal variability observed in the signal. The process described in this section is depicted in Fig. 3. The anomalies at a given time $l$, denoted by $A(l)$, are defined as follows:

$$A(l) = \frac{s(l) - \bar{s}(l)}{\sigma(l)}. \tag{4}$$

Here, $s(l)$ represents the signal at time $l$, $\bar{s}(l)$ is the mean of the signal at that time, and $\sigma(l)$ refers to its standard deviation. In this context, the time variable $l$ denotes the specific day of the year, i.e., the third of March, and is determined based on a multi-year mean. The normalization process ensures that the signal exhibits a zero mean, which facilitates identifying extreme events as data points that fall outside a specified distribution range.

After normalizing the data and delineating the anomalies, we characterize extreme events as data points falling outside a specific quantile, chosen to be 0.90, 0.91, ..., or 0.99. Based on the selected quantile, we define two threshold parameters $\kappa_+$ and $\kappa_-$ to represent positive and negative extremes, respectively. We determine these parameters individually for each site involved in our study.





During the training phase, which spans 2000 to 2014, we determine the threshold values $\kappa_+$ and $\kappa_-$. We apply these threshold
values to the test dataset comprising the years from 2014 to 2020 to determine the extreme events in this latter dataset. The
normalization of the data in the test datasets is done with the mean and SD values obtained from the training dataset.

## 2.6    Metrics

### 2.6.1    General Metrics

In this study, we evaluated the predictive accuracy of our model using two primary metrics: the normalized root mean square
error (NRMSE) and the symmetric mean absolute percentage error (SMAPE).

The NRMSE is derived from the root mean square error (RMSE) (Hyndman and Koehler, 2006), and is defined as:

$$\text{NRMSE} = \sqrt{\frac{\sum_{t=1}^{n}(\tilde{v}(t) - v(t))^2}{N}} \frac{1}{v_{\max} - v_{\min}}, \tag{5}$$

where $v(t)$ represents the observed target variable at the $t$-th observation among a total of $N$ observations, and $\tilde{v}(t)$ is the
corresponding model prediction. To facilitate comparisons across different sites, we normalized this metric using the range of
the observed data, which is computed as the difference between the maximum $v_{\max}$ and minimum $v_{\min}$ observed values, with
$t \in \{1, ..., n\}$.

In addition to NRMSE, we use SMAPE to assess the predictive performance of our model. SMAPE, which is a dimension-
agnostic measure, is given by the formula

$$\text{SMAPE} = \frac{100}{n} \sum_{t=1}^{n} \frac{|v(t) - \tilde{v}|}{|v(t)| + |\tilde{v}|}, \tag{6}$$

where $n$ denotes the number of observations, this metric was chosen based on its ability to provide a symmetric measurement
of the absolute percentage error, thereby affording a balanced view of the forecast accuracy (Makridakis, 1993; Hyndman and
Koehler, 2006).

### 2.6.2    Entropy-Complexity Plots

In this study, we also use information theory-based quantifiers to analyze the model's residuals, defined as the differences
between the model's prediction and the actual measurements. Based on the approach proposed by Sippel et al. (2016), we
employ entropy-complexity (EC) plots. These plots return a visual representation of the amount of information still present in
the residuals. Higher information content would indicate that the models do not sufficiently approximate the target variable. On
the other hand, values of EC closer to white noise would suggest that the models fully reproduce the target variable's behavior.
We illustrate this approach, closely following the exposition provided by Sippel et al. (2016).
To generate the EC plots, we consider a metric $\mathcal{H}[P]$ of a probability distribution $P = \{p_i; i = 1, ..., N\}$, with $N$ possible
states and with $\sum_{i=1}^{N} p_i = 1$, to quantify the information content of a time series. One such metric is the Shannon entropy $\mathcal{S}[P]$,



$$S[P] = -\sum_{i=1}^{N} p_i \ln[p_i], \qquad (7)$$

which is maximized $S[P_e] = S_{\max} = \ln N$ for the uniform distribution $P_e = \{p_i = \frac{1}{N}; \forall i = 1, \ldots, N\}$. We can then define the normalized entropy

$$\mathcal{H}[P] = \frac{S[P]}{S_{\max}}, \qquad (8)$$

which is the first metric used in the EC plots. In addition to the information content of a time series, we are interested in quantifying the complexity $\mathcal{C}[P]$. Following Lopez-Ruiz et al. (1995), we use a definition of complexity $\mathcal{C}[P]$, which is the product of a measure of information, such as entropy $\mathcal{H}[P]$, and a measure of disequilibrium $\mathcal{Q}_J[P, P_e]$.

$$\mathcal{C}[P] = \mathcal{Q}_J[P, P_e]\mathcal{H}[P], \qquad (9)$$

In this context, disequilibrium takes the meaning of distance from the uniform distribution of the available states of a system. The definition of disequilibrium $\mathcal{Q}_J[P, P_e]$ makes use of the Jensen-Shannon divergence $\mathcal{J}[P, P_e]$, which quantifies the difference between probability distributions (Grosse et al., 2002), and it is defined as

$$\mathcal{J}[P, P_e] = \mathcal{Q}_0 \left\{ S\left[\frac{P + P_e}{2}\right] - \frac{1}{2}(S[P] - S[P_e]) \right\}, \qquad (10)$$

where $\mathcal{Q}_0$ is a normalization constant, (Lamberti et al., 2004; Rosso et al., 2007), chosen such that $\mathcal{Q}_J[P, P_e] \in [0, 1]$.

The computations of entropy and complexity rely on the probability distribution associated with the data. To determine this distribution, we leverage the method proposed by Bandt and Pompe (2002), which analyzes time series data by comparing neighboring values. It involves dividing the data into a set of patterns based on the order of the values and then calculating the probability of each pattern occurring (Rosso and Masoller, 2009). The pattern separation is obtained by embedding the time series in a $D$ dimensional space with a time lag $\tau$. We set D = 6 and $\tau = 1$ as proposed by (Rosso et al., 2007; Sippel et al., 2016). The complexity measure's theoretical upper and lower bounds can now be computed (Martin et al., 2006) and are shown in the plots.

The calculations of the EC plots were performed with the Julia package `ComplexityMeasures.jl` (Haaga and Datseris, 2023) from `DynamicalSystems.jl` (Datseris, 2018).

### 2.6.3 Extremes as Binary Events

To analyze extreme events, we set thresholds $\kappa_+$ and $\kappa_-$, as described earlier (in Sect. 2.5) to identify values as either extremes or not extremes. We adopt this binary approach for both the observed data and the data predicted by the models.

Following the method outlined by Hogan and Mason (2011), we classify the outcomes into four categories: hits $a$, where the model correctly identifies an extreme event; false alarms $b$, where the model incorrectly flags a value as an extreme event; misses $c$, where the model fails to identify an extreme event; and correct rejections $d$, where the model correctly identifies a non-extreme event.





Given $n$ observation points, we employ the following metrics to assess the model's performance on detecting extremes, (Barnes et al., 2009). We define the probability of detection POD as the ratio of correctly identified extreme events and the total number of extreme events, the probability of false detection POFD as the ratio of incorrectly flagged extreme events and all events that are not extreme, the probability of false alarm POFA as the ratio of false alarms and all predicted extreme events, and the proportion correct PC as the ratio of all accurate predictions (both hits and correct rejections) and the total number of observations, given by

$$POD = \frac{a}{a+c}, \; POFD = \frac{b}{b+d}, \; POFA = \frac{b}{a+b}, \; PC = \frac{a+d}{n}, \tag{11}$$

respectively.

These metrics are standard tools for evaluating deep learning models for predicting atmospheric variables such as wind speed (Scheepens et al., 2023) and precipitation (Shi et al., 2015).

## 3 Results

In the following, we first compare the different models by standard evaluation metrics such as the normalized root mean squared error, NRMSE, and the symmetric mean absolute percentage error, SMAPE (Sect. 3.1). Second, we extend the comparison to information-theoretic quantifiers in the entropy-complexity (EC) plane (Sect. 3.2). We perform this comparison for two cases: (i) the full time series (FS), which encompasses all data points in the year, and (ii) the meteorological growing season (GS), which encompasses the months between May and September only. This division helps us differentiate the models' ability to capture the seasonal cycle, dominated by an oscillation, from their performance in the more stable growing season conditions, where more minor variations are likely harder to be represented. Last, we focus on extreme events and compare the models' performance on extreme events in 3.3.

### 3.1 Comparison of Prediction Accuracy

We present our results in Table 1, showing that the ESN outperformed all other models for both the FS and GS signals, closely followed by the LSTM. While the GRU exhibited comparable results to the LSTM for the FS signal, the differences became more pronounced in the GS signal. In contrast, the RNN consistently delivered the least favorable results across the board and exhibited the highest standard deviation (SD) among the analyzed models. In Fig. A1, we include the comparison of these metrics per site, which shows great variation in the models' performance across different sites.

The performance rankings of the models remained consistent when transitioning from the FS signal to the GS signal. Notably, the SMAPE metric indicated increased accuracy for all models, while the NRMSE metric suggested decreased accuracy. Similarly, we observe a reduction in the SD for the SMAPE but an increase in the NRMSE. This increase is very noticeable in the GRU and RNN models.

The models generally exhibited similar performance, with the ESN yielding slightly better results and the RNN demonstrating the least accurate forecasts. The gated methods showed similar performances. While these metrics provide an initial picture





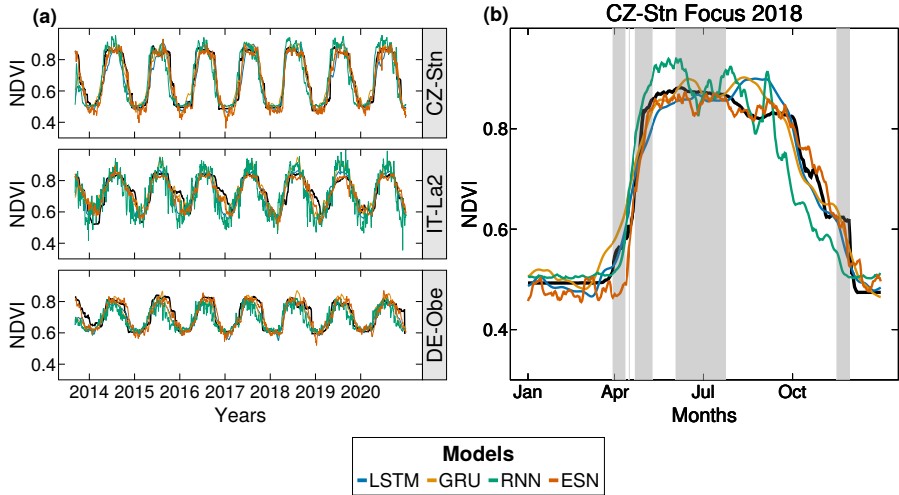

**Figure 4. Time series and predictions for selected locations.** This figure illustrates the predictive performance of four distinct recurrent architectures at specified NDVI time-series locations. The baseline is shown in black, while the predictions only use a singular run from a set of 50 per model. Panel **(a)** delineates the results obtained at three selected locations. Subsequently, panel **(b)** offers a magnified view of the outcomes at the CZ-Stn location in 2018, highlighting the extremes defined by a 90% threshold. It is pertinent to underline that the predictions generated by the RNNs carry considerable noise, thereby affecting the interpretations drawn from the entropy-complexity plots.

of the models' performance in the task, the results do not indicate a clear best performer among the models. These similarities in the results' metrics emphasize the need for further exploration using alternative evaluation tools.

**Table 1. Accuracy of the models**. The table illustrates the performance of the four different models across various scenarios. "FS" represents the "full season," representing the full data set used for predictions without isolating specific events or months. "GS" denotes the "growing season," which encompasses the peak summer months in addition to May and September. Models with the highest accuracy in each category are highlighted using bold text. The arrows pointing down (↓) near the metric's name indicate that smaller values represent higher accuracy.

| | | LSTM | GRU | RNN | ESN |
|---|---|---|---|---|---|
| **FS** | SMAPE ↓ | $5.97 \pm 1.84$ | $6.19 \pm 2.23$ | $7.86 \pm 2.69$ | $\mathbf{4.77 \pm 1.14}$ |
| | NRMSE ↓ | $0.172 \pm 3.23 \cdot 10^{-2}$ | $0.170 \pm 3.68 \cdot 10^{-2}$ | $0.218 \pm 4.24 \cdot 10^{-2}$ | $\mathbf{0.135 \pm 3.29 \cdot 10^{-2}}$ |
| **GS** | SMAPE ↓ | $4.57 \pm 1.25$ | $4.84 \pm 1.65$ | $6.57 \pm 1.89$ | $\mathbf{3.14 \pm 0.82}$ |
| | NRMSE ↓ | $0.222 \pm 7.04 \cdot 10^{-2}$ | $0.229 \pm 9.18 \cdot 10^{-2}$ | $0.297 \pm 10.61 \cdot 10^{-2}$ | $\mathbf{0.153 \pm 5.19 \cdot 10^{-2}}$ |





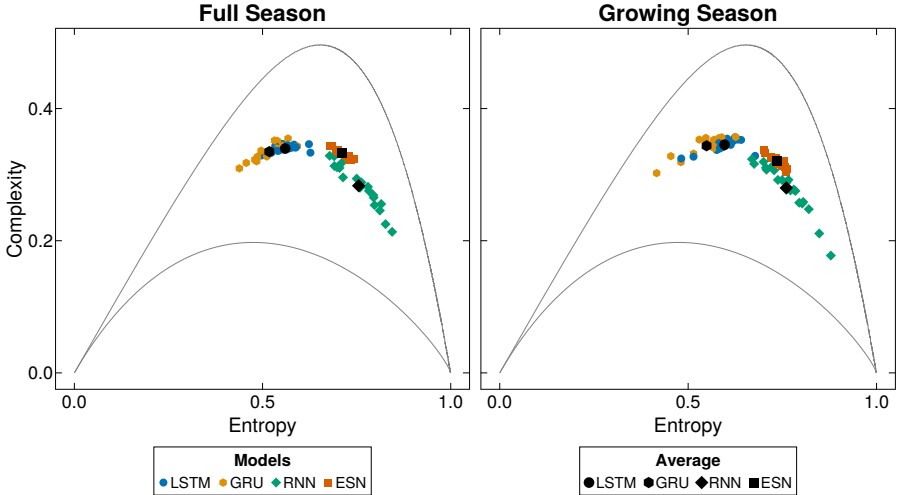

**Figure 5. Entropy-complexity curves of model's residuals**. The entropy-complexity plots are computed from the residuals of each model at each location (color). The mean of each model's performance over all locations is shown in black. These plots visualize the amount of information and complexity left in the residuals. Ideal values would reside in the lower left corner, symbolizing white noise in the residuals. The gray lines denote the theoretical upper and lower bounds.

### 3.2 Comparison of the Entropy-Complexity

Drawing from information theory, we use entropy-complexity (EC) plots (introduced in Sec 2.6.2) to examine the residuals, defined as the differences between the model's predictions and actual measurements. Residuals convey valuable information about a model's performance. In an ideal scenario, where a model perfectly represents system dynamics, these residuals should resemble white noise and position in the lower right corner of the EC plane (Sippel et al., 2016).

Figure 5 shows the EC plots of the models' residuals. Additionally, we show the mean of these metrics per model architecture across all sites. We find that the residuals cluster by model across all locations, with minimal overlap between each model's clusters. The LSTM and GRU models occupy the curve's ascending left side, indicating the presence of more structure in the residuals. In contrast, the ESN and, to a greater extent, the RNN are positioned closer to the white noise region, implying less structure in the residuals.

Comparing the FS signal to the GS signal shows no apparent differences in the results, underscoring the consistency of model performance. Based on the positioning of RNNs in the EC plots, we would expect a substantially improved performance of these models compared to the other model architectures. However, inspecting the predicted time series suggests a different explanation: the RNN model predictions shows large variability (Fig. 4, which obfuscates the underlying dynamics. Because the EC plots capture the resulting residuals' noise, one can misinterpret the positioning of RNNs as favorable results.





## 3.3 Extreme Events

In this section, we use the metrics introduced in Sect. 2.6.3 to evaluate the models' ability to capture extreme vegetation dynamics. Fig. 6a shows a distinct division in the probability of detection (POD): the LSTM and ESN models exhibit low values, indicating an inability to detect extreme events. The RNN and GRU models show higher values, indicating better performance. However, the POD values are generally low for all models. Furthermore, all models display some level of overlap in the confidence bands. Finally, it is possible to observe a worsening of the results with increased quantiles. Figure 6b shows a lower probability of false detection (POFD) of ESNs, indicating that they perform better at avoiding the prediction of extremes when there are none compared to the other models. Conversely, the RNN, which predicts noisy behavior, as shown in Sect. 3.1, exhibits higher POFD values. The probability of false alarm (POFA; Fig. 6c) follows a similar pattern, with the ESN outperforming the other models. The GRU and LSTM display more comparable behaviors, while the RNN lags behind, showing the least favorable performance among all models. The displayed metric consistently leans towards the higher end of the potential value range. Like in the case of the POD, the values of the POFA get worse with each increase of the quantiles. Finally, Fig. 6d shows the probability of correctness (PC). The ESN leads the graph despite showing the lowest POD. Following closely are the LSTM and GRU, which exhibit similar performance. The RNN maintains the poorest performance across all metrics, emphasizing its limitations in capturing extreme vegetation dynamics.

To examine the prediction performance of extreme NDVI reduction during the summer season, we focus on June, July, and August using only $\kappa_-$, showcased in Fig. 7. In Fig. 7a, we see the probability of detection (POD) for different models. The RNN performs best, outperforming the LSTM, GRU, and ESN, delivering similar results. Figure 7b shows the probability of false detection (POFD). Here, the RNN performs the worst, while the ESN stands out with the best values and the lowest standard deviation (SD) among all models. The gating-based models, LSTM and GRU, perform almost identically. We find similar performances among models in Fig. 7c. The ESN demonstrates a decrease in the probability of false alarm (POFA) values for the 0.92 quantiles. Any small gaps in performance observed for lower quantiles are effectively bridged at higher values. Finally, Fig. 7d illustrates the ESN as the top-performing model, exhibiting the lowest SD among the models. The gated models, LSTM and GRU, display similar performances, while the RNN ranks as the poorest-performing model in this analysis of NDVI summertime decreases.





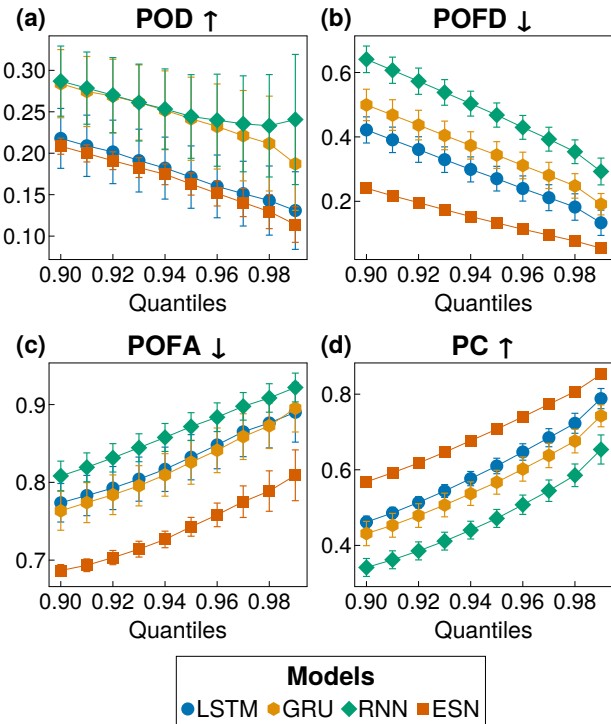

**Figure 6. Extremes as binary events.** The results of quantiles spanning from 0.90 to 0.99 are shown. All values are expressed as percentages. In **(a)**, we show the detection percentage (POD), indicating how many extreme events have been detected. In **(b)**, we show the percentage of false detection (POFD). In **(c)** we depict the probability of false alarms (POFA). Finally, in **(d)** we show the proportion correct (PC). The error bars represent the standard deviation over 50 simulations with different initializations for each model. The arrows (up ↑, and down ↓) by the metric's name indicate the direction of the optimal values.





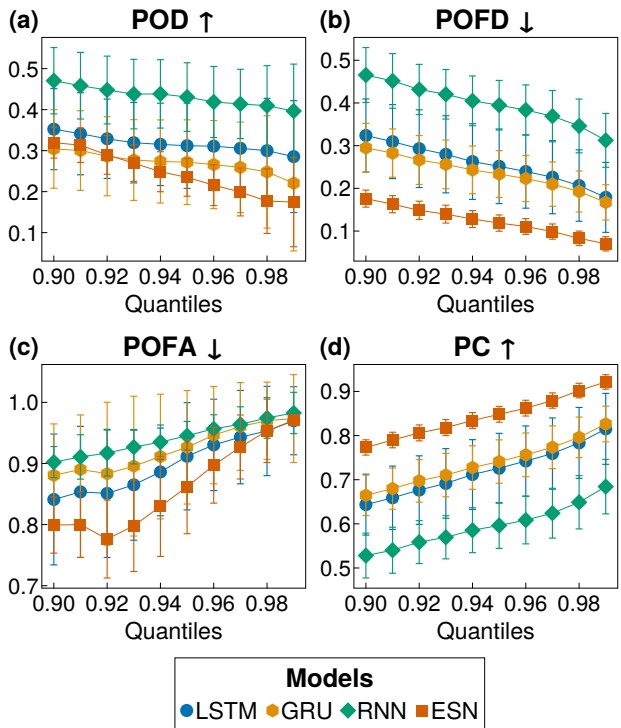

**Figure 7. Negative extreme events for summer months.** Only the negative extremes during the summer period are considered here. The results of quantiles spanning from 0.90 to 0.99 are shown. All values are expressed as percentages. In **(a)**, we show the detection percentage (POD), indicating how many extreme events have been detected. In **(b)**, we show the percentage of false detection (POFD). In **(c)** we depict the probability of false alarms (POFA). Finally, **(d)** we shown the proportion correct (PC). The error bars represent the standard deviation over 50 simulations with different initializations for each model. The arrows (up ↑, and down ↓) by the metric's name indicate the direction of the optimal values.





## 4 Discussion

In the following section, we discuss the results, starting in 4.1 with the performance of the models and their comparison in modeling vegetation greenness in response to climate drivers. Subsequently, in 4.2, we analyze their performance in learning the extremes in the normalized vegetation index (NDVI) time series. Finally, we discuss the limitations of this study and highlight future research directions in 4.3.

### 4.1  Model's Performance and Comparison

Similar to previous investigations (Benson et al., 2023), this work underlines the ability of recurrent neural networks to model vegetation greenness. This is highlighted by the good values of NRMSE and SMAPE obtained by the ESNs and LSTMs. Comparing models, ESNs perform better in standard measures, but their NRMSE and SMAPE standard deviations (SD) overlap with those of LSTMs and GRUs. Furthermore, RNNs ranked last in this comparison. Based on these metrics, gated architectures and ESNs appear almost equally effective, with ESNs having a slight advantage. The performance varies considerably across sites, as illustrated in Fig. A1, and can explain the overlap in the standard deviation. Despite site specific training, this difference in performances still persists, and it could be the subject of future studies.

To extend the comparison, we also provide a comparison of entropy-complexity (EC) plots. The EC plots showed the presence of fewer leftover signals in the residuals of the ESNs' predictions. The LSTMs followed, with the GRUs showing the worst performance. Contrary to the previous metrics, RNNs seem to outperform all other models in the EC plots. However, examining their predictions shows this is not the case due to the high noise in RNN predictions, distorting the EC plot insights. Thus, our results illustrate that it is crucial not to rely solely on a single set of metrics when evaluating results in similar studies.

Existing studies center on creating a single global model (Kraft et al., 2019; Diaconu et al., 2022). However, we opted to train individual models for each site. Our goal was to examine how effectively each architecture could capture the specific dynamics of an ecosystem without giving the models any additional contextual information. This approach emphasizes the models' inherent capability to understand each ecosystem's unique attributes.

### 4.2  Do the Models Capture Extremes?

Our results for the analysis of vegetation extreme responses to climate drivers using daily data showed that none of the models notably outperformed the others in all the metrics considered. The ESNs performed well in avoiding false alarms but failed to obtain a high accuracy for predicting actual extremes, performing similarly to LSTMs. ESNs also showed the lowest standard deviation among all the models considered. RNNs performed worst among all metrics analyzed. Overall, all the models showed poor results for this task. When we only considered negative extremes in the summer season, the results showed again that the recurrent architectures could not correctly capture extremes in vegetation responses to climate forcing. The ESNs performed best in avoiding false extremes and the overall accuracy of the task's execution, while the RNNs showed the worst prediction performance.





Motivated by recent ESN applications to data from laboratory experiments (Pammi et al., 2023), we expected ESNs to outperform the other models in the extreme prediction task. Instead, the results showed that, while the ESN provided good results, they were not outstanding compared to the BPTT gated models. Additionally, none of the models showed good results, contrary to other applications in the Earth sciences showcasing the successful application of recurrent models to extremes

(Frame et al., 2022).

## 4.3  Limitation and Future Directions

The study is subject to a few limitations. The most crucial limitation pertains to data quality. The provided NDVI data undergoes gap-filling, employing state-of-the-art techniques; nevertheless, certain artifacts persist in the final output. Another limitation arises from the availability of daily data solely spanning the last 21 years. This constraint hampers the training of machine

learning models, notorious for their dependency on abundant data points for optimal performance. Marcolongo et al. (2022) show that a deep learning approach is suitable for detecting extreme vegetation responses. Using simulations, the study had access to 100,000 years of daily data, indicating that if enough data is available, neural networks can properly model extremes. Furthermore, the metrics we used were shown to be sensitive to factors like noise. Using various metric sets, we identified limitations in methods like the EC plots. For instance, the noise in the RNN's prediction was incorrectly attributed to residual

noise, distorting the outcomes. Moreover, standard metrics like NRMSE and SMAPE provide a narrowed view. This is evident when comparing ESNs and LSTMs: even though they had similar scores with these metrics, their distinctions were clear in the EC plots.

ESN and LSTM demonstrated comparable performance in modeling vegetation greenness. However, these models exhibited limited predictive capability for extreme events. Spatial information has proven helpful in similar investigations (Requena-

Mesa et al., 2021; Diaconu et al., 2022; Kladny et al., 2022; Robin et al., 2022; Benson et al., 2023) and it could be beneficial to explore the extent to which it contributes to extreme conditions. Furthermore, the models used could be tailored more to the task. In (Bonavita et al., 2023), the authors point out the limitations in using the mean-squared-error loss function, a practice still widely diffused in the field and adopted in this paper. Furthermore, Rudy and Sapsis (2023) show the superior performance of loss functions based on weighting and relative entropy compared to other loss functions in learning extreme events. Finally,

further investigations could benefit from deeper explorations of ESNs. With their user-friendly nature, straightforward hyper-parameter tuning, and fast training, they stand out as a robust modeling method, able to compete with top-tier deep learning architectures such as LSTMs.

## 5  Conclusions

We compared the performance of recurrent neural networks in modeling biosphere dynamics, i.e., vegetation states, in response

to climate drivers. Using daily data, we assessed the effectiveness of these network architectures in capturing extreme anomalies within these vegetation dynamics. To discern variations in performance across different scenarios, we employed various metrics such as normalized root mean square error and symmetric mean absolute percentage error paired with information theory



quantifiers. Our findings revealed that ESNs and LSTMs performed similarly for most analyzed metrics, indicating that no single model outperformed others. Additionally, all the models under investigation failed to model the extreme responses of

the vegetation. This work highlights the necessity to continue refining and developing specialized models that can more adeptly capture extreme vegetation responses to climate factors.

*Code availability.* The code for this study is available at https://github.com/MartinuzziFrancesco/rnn-ndvi

*Data availability.* The data used in this study is available online:

– E-OBS dataset (Cornes et al., 2018) at www.ecad.eu/download/ensembles/download.php

– FluxnetEO dataset (Walther et al., 2022) at https://meta.icos-cp.eu/collections/tEAkpU6UduMMONrFyym5-tUW

## Appendix A: Recurrent Neural Network Details

### A1    Recurrent Neural Networks

The most basic version of RNN was proposed by Elman (Elman, 1990). The equations used to obtain the hidden state $\boldsymbol{x}(t) \in \mathbb{R}^{d_x}$ can be described as follows:

$$\boldsymbol{x}(t) = \sigma(\mathbf{W}_{\text{in}}^x \boldsymbol{u}(t) + \mathbf{W}^x \boldsymbol{x}(t-1) + \boldsymbol{b}^x) \tag{A1}$$

where $\sigma$ is the activation function, $\mathbf{W}_{\text{in}}^x \in \mathbb{R}^{d_x \times d_u}$ and $\mathbf{W}^x \in \mathbb{R}^{d_x \times d_x}$ are the weight matrices and $\boldsymbol{b}^x \in \mathbb{R}_x^d$ is a bias vector. In addition, $\boldsymbol{u}(t) \in \mathbb{R}_u^d$ is the input vector at time $t$.

The main problem of this model is the vanishing and exploding gradient due to the multiple calculations of the gradient during backpropagation through time (Werbos, 1988).

### A2    Long Short Term Memory

For LSTMs the hidden state $\boldsymbol{x}(t) \in \mathbb{R}^{d_u}$ is obtained as follows (Hochreiter and Schmidhuber, 1997)

$$\boldsymbol{f}(t) = \sigma_g(\mathbf{W}_{\text{in}}^f \boldsymbol{u}(t) + \mathbf{W}^f \boldsymbol{x}(t-1) + \boldsymbol{b}^f) \tag{A2}$$

$$\boldsymbol{i}(t) = \sigma_g(\mathbf{W}_{\text{in}}^i \boldsymbol{u}(t) + \mathbf{W}^i \boldsymbol{x}(t-1) + \boldsymbol{b}^i) \tag{A3}$$

$$\boldsymbol{o}(t) = \sigma_g(\mathbf{W}_{\text{in}}^o \boldsymbol{u}(t) + \mathbf{W}^o \boldsymbol{x}(t-1) + \boldsymbol{b}^o) \tag{A4}$$

$$\tilde{\boldsymbol{c}}(t) = \tanh(\mathbf{W}_{in}^c \boldsymbol{u}(t) + \mathbf{W}^c \boldsymbol{x}(t-1) + \boldsymbol{b}^c) \tag{A5}$$

$$\boldsymbol{c}(t) = \boldsymbol{f}(t) \odot \boldsymbol{c}(t-1) + \boldsymbol{i}(t) \odot \tilde{\boldsymbol{c}}(t) \tag{A6}$$

$$\boldsymbol{x}(t) = \boldsymbol{o}(t) \odot \sigma_x(\tilde{\boldsymbol{c}}(t)) \tag{A7}$$





where $\boldsymbol{f}(t)$ is the *forget* gate, $\boldsymbol{i}(t)$ is the *input* gate and $\boldsymbol{o}(t)$ is the *output* gate. The activation functions $\sigma_g$ are usually set to be sigmoid, and $\sigma_x$ is set to be the hyperbolic tangent. However, it can be set to unity for some variations of the model (e.g.,

"peephole" LSTM, Gers and Schmidhuber, 2000). The matrices $\mathbf{W}_{\text{in}}^j \in \mathbb{R}^{d_x \times d_u}$ and $\mathbf{W}^j \in \mathbb{R}^{d_x \times d_x}$ for $j \in \{f,i,o,c\}$ are the weight matrices while the vectors $\boldsymbol{b}^j \in \mathbb{R}^{d_x}$ for $j \in \{f,i,o,c\}$ are bias vectors. The vector $\boldsymbol{u}(t) \in \mathbb{R}^{d_u}$ represents the input vector at time $t$.

### A3    Gated Recurrent Units

The equations to obtain the hidden state $\boldsymbol{x}(t) \in \mathbb{R}^{d_x}$ for GRUs are defined as follows (Cho et al., 2014):

$$\boldsymbol{r}(t) = \sigma(\mathbf{W}_{\text{in}}^r \boldsymbol{u}(t) + \mathbf{W}^r \boldsymbol{x}(t-1) + \boldsymbol{b}^r) \tag{A8}$$

$$\boldsymbol{z}(t) = \sigma(\mathbf{W}_{\text{in}}^z \boldsymbol{u}(t) + \mathbf{W}^z \boldsymbol{x}(t-1) + \boldsymbol{b}^z) \tag{A9}$$

$$\tilde{\boldsymbol{x}}(t) = \tanh(\mathbf{W}_{in}^x \boldsymbol{u}(t) + \mathbf{W}^x(\boldsymbol{r}(t) \odot \boldsymbol{x}(t-1)) + \boldsymbol{b}) \tag{A10}$$

$$\boldsymbol{x}(t) = \boldsymbol{z}(t) \odot \boldsymbol{x}(t-1) + (1 - \boldsymbol{z}(t)) \odot \tilde{\boldsymbol{x}}(t) \tag{A11}$$

where $\boldsymbol{r}(t)$ is the *reset* gate, $\boldsymbol{z}(t)$ is the *update* gate and $\boldsymbol{u}(t) \in \mathbb{R}^{d_u}$ is the input signal. The activation functions $\sigma$ are set to be

sigmoid. As in the LSTM, the matrices $\mathbf{W}_{\text{in}}^j \in \mathbb{R}^{d_x \times d_u}$ and $\mathbf{W}^j \in \mathbb{R}^{d_x \times d_x}$ for $j \in \{r,z,x\}$ are the weight matrices while the vectors $\boldsymbol{b}^j \in \mathbb{R}^{d_x}$ for $j \in \{r,z,x\}$ are bias vectors.

### A4    Echo State Networks

The hidden state $\boldsymbol{x}(t) \in \mathbb{R}^{d_x}$ for the ESN is defined as (Jaeger, 2001):

$$\boldsymbol{x}(t) = (1 - \alpha)\boldsymbol{x}(t-1) + \alpha\tanh(\mathbf{W}_{\text{in}}\boldsymbol{u}(t) + \mathbf{W}\boldsymbol{x}(t-1)), \tag{A12}$$

where $\alpha$ is the leaky coefficient and $\boldsymbol{u}(t) \in \mathbb{R}^{d_u}$ is the input data. Similarly as before the matrices $\mathbf{W}_{\text{in}} \in \mathbb{R}^{d_x \times d_u}$ and $\mathbf{W} \in \mathbb{R}^{d_x \times d_x}$ are the weights matrices, with the difference that this time these matrices do not undergo training or change. Since they are kept fixed, the initialization of these matrices also plays a role in predicting the model. The standard choices are to create $\mathbf{W}_{\text{in}}$, also called *input* matrix, as a dense matrix with weights randomly sampled from a uniform distribution in the range $[-\sigma, \sigma]$. The weight matrix $\mathbf{W}$ is usually referred to as the *reservoir* matrix, and it is usually built from an Erd "os–Rényi graph

configuration. This matrix shows a high sparsity, usually in the $1 - 10\%$ range, and its values are also randomly sampled from a uniform distribution $\in [-1, 1]$. Subsequently, the matrix is scaled to obtain a chosen spectral radius $\rho(\mathbf{W})$. The values of the spectral radius, size of the matrix, and its sparsity are the main hyperparameters for ESN models.

The output is obtained through a linear feed-forward layer:

$$\tilde{\boldsymbol{v}}(t) = \mathbf{W}_{\text{out}}\boldsymbol{x}(t), \tag{A13}$$

where $\mathbf{W}_{\text{out}} \in \mathbb{R}^{d_v \times d_x}$ is the *output* matrix. This matrix is the only one whose weights undergo training. Unlike the models illustrated before, this training is not done using BPTT but simple linear regression. During the training phase, all the inputs



$\boldsymbol{u}(t) \in \mathbb{R}^{d_u}$ are passed through the ESN, and the respective expansions (hidden states) are saved column-wise in a *states* matrix $\mathbf{X} \in \mathbb{R}^{d_x \times d_T}$ where $T$ is the length of the training set $t = 1, ..., T$. In a similar way, the matrix $\mathbf{Y}^{\text{target}} \in \mathbb{R}^{d_v \times d_T}$ is built with the desired output $\boldsymbol{v} \in \mathbb{R}^d_v$ is stacked column-wise. This way, the output layer can be obtained using ridge regression with the

following closed form:

$$\mathbf{W}_{\text{out}} = \mathbf{Y}^{\text{target}} \mathbf{X}^{\text{T}} (\mathbf{X}\mathbf{X}^{\text{T}} + \beta\mathbf{I})^{-1}, \tag{A14}$$

where $\mathbf{I}$ is the identity matrix and $\beta$ is a regularization coefficient.

### Appendix B: Computational Details

### B1  Experimental Settings

All the figures in this paper (with the exception of Fig. 1) have been obtained using the Julia language (Bezanson et al., 2017) package `Makie.jl` (Danisch and Krumbiegel, 2021). All the models have been optimized using grid search. For each optimal set of hyperparameters, 100 different runs have been carried out for each model and site. The 50 best-performing of these runs have been used for the results shown here. Unless specified otherwise, the results showcased are mean and standard deviation over these 50 runs per location, over all locations.

### B2  Details of the models

In the proposed models based on TBPTT, the parameters are optimized using the stochastic optimization method called Adam (Kingma and Ba, 2014), while dropout (Srivastava et al., 2014) is used to protect against over-fitting. Additionally, early stopping is employed to halt training when the validation loss has not changed, with a patience factor of 50 epochs. The weights $\theta_W$ are initialized using the technique proposed in (Glorot and Bengio, 2010), drawing from a uniform distribution.

The biases $\theta_b$ are initialized by drawing values from a uniform distribution.

Different layers of recurrent networks are also stacked on top of each other, providing a deeper model. The number of layer and weights per layer is also optimized. All the hyper-parameters undergo optimization using grid search using the root mean square error (RMSE) as a guiding measure. Split temporal cross-validation is used (Cerqueira et al., 2020).

### B3  Grid Search Parameters

Table A1 provides the parameters used for the grid search in this study. The values are either given as a list, divided by a comma $(a, b, c)$, or as an interval, separated by a colon $(start : step : stop)$. In the first case, the values indicated are the values used. In the second case, the values used are between the first and last, with a step size indicated by the second value.

### B1  Site Comparison

Here, we provide a comparison of the models' performance across the study sites.



**Table A1. ESNs hyperparameters grid search values.**

| ESN | | | |
|---|---|---|---|
| Sparsity | Ridge coefficient | Leaky coeff. | Radius |
| $0.01 : 0.01 : 0.1$ | $1.0 \times 10^{-2}, 10^{-3}, 10^{-4}, 10^{-5}$ | $0.5 : 0.05 : 1.0$ | $0.9 : 0.05 : 1.5$ |

**Table B1. RNN/LSTM/GRU hyperparameters grid search values.**

| RNN/LSTM/GRU | | | |
|---|---|---|---|
| Learning rate | Hidden dimension | Num. layers | Dropout |
| $1.0 \times 10^{-3}, 10^{-4}, 10^{-5}$ | $32, 64, 128$ | $2, 3, 4$ | $0.1 : 0.1 : 0.4$ |

*Author contributions.* FM and KM conceptualized the work, and FM carried out the simulations. KM, MM, GCV, and TW provided suggestions for the analysis. KM and MM supervised the work. DM formulated the data pre-processing pipeline, while FM implemented it. FM wrote the manuscript with contributions from all authors.

*Competing interests.* The authors declare that there are no competing interests.

*Acknowledgements.* We thank Sophia Walther for explaining the FluxnetEO data set in detail. This research was supported by grants from
the European Space Agency and ESA (AI4Science - Deep Extremes and Deep Earth System Data Lab). FM and MDM acknowledge the financial support from the Federal Ministry of Education and Research of Germany and by Sächsische Staatsministerium für Wissenschaft, Kultur und Tourismus in the programme Center of Excellence for AI-research "Center for Scalable Data Analytics and Artificial Intelligence Dresden/Leipzig", project identification number: ScaDS.AI. KM and MDM acknowledge support from the Saxon State Ministry for Science, Culture and Tourism (SMWK project 232171353). DM and MDM acknowledge support from the "Digital Forest" project, Ministry of Lower-
Saxony for Science and Culture (MWK) via the program Niedersächsisches Vorab (ZN3679); MDM acknowledges support from the German Aerospace Center, DLR (ML4Earth). We thank the European Union for funding XAIDA via Horizon 2020 grant no. 101003469.



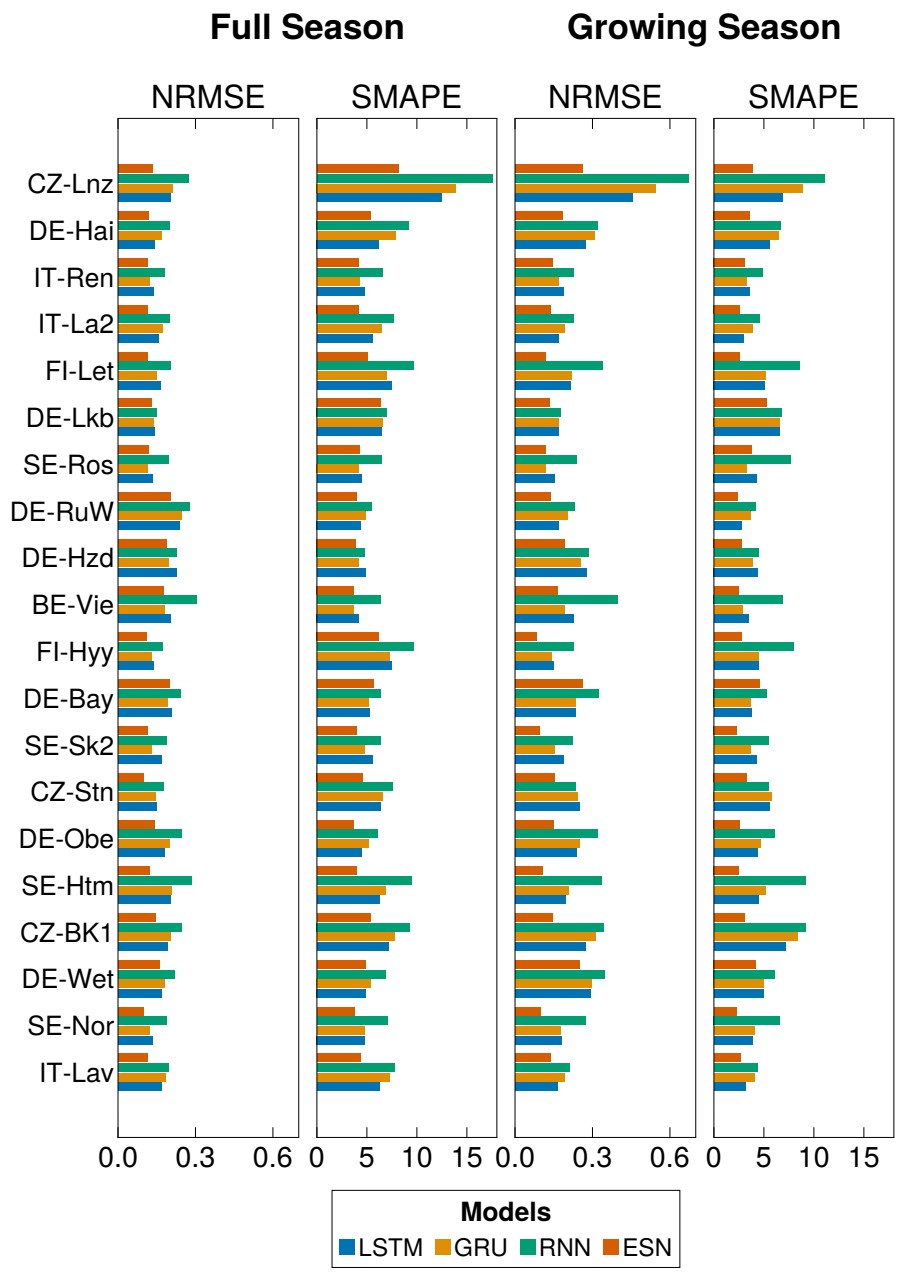

**Figure A1. Mean results across locations**. We show the metrics for all the analyzed locations and all models. Full season refers to the use of the full dataset for the results. Growing season indicates that we only utilized the months between May and September (included). The figure shows the mean of 50 runs per location.



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
