# Peer review of "Learning Vegetation Response to Climate Drivers and Extremes with Recurrent Neural Networks"

_EGUsphere, 2023_

## Author Comment (AC1)

May 21, 2024

Nonlinear Processes in Geophysics
Manuscript ID: EGUSPHERE-2023-2368

Dear Referee 1,

We thank you for taking the time to read our manuscript and for the helpful comments posted. This letter details the changes we made to address the points raised.

We highlight the referee comments in *blue italics*. We provide our replies to the comments and the changes we made to the manuscript. Additionally, we showcase the manuscript additions and alterations in **bold font**.

1. *Title. Considering that the models applied to all NDVI time series and extreme cases are analyzed for performance comparison, I would suggest removing "extreme" from the title to better reflect the broader application of the model*

   While we wanted the title to reflect a specific section of the analysis, given its novelty in the field, we now see how the wording could be misleading. We suggest the following alternative title:

   **Learning Vegetation Response to Climate Drivers and Extremes with Recurrent Neural Networks**

   While the new title maintains the word "extreme," we shifted the focus to underline the study's broader scope.

2. *Section 2.1. The models are trained separately for each site. Is it feasible to train a universal model with all the stations by incorporating some static features (such as elevation, latitude, longitude, etc.,)? There have been studies showing the LSTM model benefits from diverse training datasets. How might this approach impact conclusions, especially regarding whether gated models benefit more from augmented training data given their complex architectures?*

   Thank you for the interesting discussion point. It is feasible to train a global model over all the investigated locations even without incorporating static features. We presume that the global model would show slightly worse overall performances compared to what we showed in the paper. This is likely due to the complexity that arises from the much larger feature space. Proper hyperparameter search and optimization would be computationally expensive and time-consuming. Consequently, performances might not be optimal for all locations. While NRMSE and SMAPE could be similar, entropy complexity plots and extremes would show a decreased performance.

   There are two reasons why additional static features would not be very helpful in this setting:

   - We used 20 locations in the study. Therefore, static variables such as elevation, latitude, and longitude do not provide meaningful insights into the model given that each point is a unique feature.

   - Our task is time series prediction in known locations. Static variables are helpful in situations in which this task is performed in unseen locations [1].

   Considering a scenario in which we have access to a much larger set of locations, the conclusions of the previous section would still apply. The proposed global model would likely perform better with static variables if given a sufficient number of locations. The additional information would come at the cost of an additional increase in the dimensionality of

the feature space, incurring an increase in the complexity of the training. Finally, to the best of our knowledge, we are not familiar with studies using ESNs as a global model, or incorporating static variables.

We appreciate your interest in understanding our modeling choices. Our study focuses on the local response of vegetation to climate drivers. To more closely investigate the models' ability in this specific setting, we used local models instead of a global approach. Thus, a global approach was beyond the scope of our study.

3. *DL features. The input features need to be clarified, particularly mean temperature and sea level pressure. Are these area means over the European continent, or are they in-situ measurements? If the latter, how do the gridded dataset correlate with in-situ NDVI values? Additionally, clarify the input time window size for the model.*

We obtained the mean temperature and sea level pressure data from the E-OBS dataset v26.0e, as mentioned in our manuscript in Section 2.1 Line 136. These climate variables are not area means over the entire European continent; they are based on in-situ measurements spatially interpolated to cover most of the European continent, as detailed in Line 137. The interpolation is twofold: (1) deterministic modeling is employed to capture the long-term spatial trend in the data for daily values. (2) Subsequently, stochastic interpolation, utilizing a Gaussian Random Field (GRF) simulation, is applied to the residuals from this model to generate the daily ensemble [2]. In this instance, the ensemble refers to an approach in which the parameters are varied within a range, the mean of which is the 'best guess.' This allows us to generate a more robust gridded dataset than a classical interpolation approach with a resolution. The E-OBS dataset has a spatial resolution of 0.1 degrees (approximately 11.1 km × 11.1 km), which enables us to approximate the climate conditions at a regional scale around each study site.

Regarding the correlation of these gridded dataset variables with in-situ NDVI values, the NDVI data was obtained from the moderate resolution imaging spectroradiometer (MODIS) in the FluxnetEO dataset v1.0, which provides high-resolution temporal and spatial data specifically for the locations of eddy covariance (EC) towers. The spatial resolution of our NDVI data cubes is 500 m, which is significantly finer than the E-OBS climate data. We employed a spatial aggregation method to ensure a meaningful comparison and correlation between these datasets. This method involved averaging the NDVI data over the whole spatial dimension available in the cube (a 3 km × 3 km area centered on the EC tower). This process is illustrated in Lines 123-125, as well as in Figure 1b and 1c. This preprocessing created a more direct and relevant comparison between the climate variables and the NDVI measurements at each study site.

Finally, regarding the input time window size for the model, the NDVI data and the climate variables (air temperature, mean sea level pressure, global radiation, and precipitation) have a daily temporal resolution and span the time period of 2000 to 2020. For our machine learning model, we used these daily measurements as input features without embeddings or fixed lookback windows, as detailed in Line 149, allowing the model to capture the day-to-day variations under the climate conditions and their impact on the NDVI measurements.

4. *Figure2, I believe subfigure a is for conventional RNN, LSTM, and b is for ESN. The figure caption is mismatched.*

Thank you for pointing out this oversight, which is now corrected. The new caption refers to the schematics as a) backpropagation-based recurrent neural networks and b) echo state networks. We propose the following new caption:

**Diagram (a) presents the conventional approach used by RNNs, GRUs, and LSTMs, involving backpropagation through time (BPTT). In contrast, diagram (b) portrays the training methodology of ESNs. The initial two layers are randomly generated and**

**remain untrained, while only the final layer undergoes a one-shot training via linear regression (LR)**

5. *Figure4, is the black line for the real/target value? I would suggest adding it in the legend.*

Thank you for pointing this out; we have updated the figure's legend to indicate that the black line represents the real/target values. We have also added a description for the greyed-out areas in the legend, which correspond to extreme events as identified by the procedure outlined in section 2.5. Furthermore, we clarified the figure caption, which is as follows:

**The target time series is shown in black, while the predictions only use a singular run from a set of 50 per model. Panel (a) delineates the results obtained at three selected locations: Germany, Italy, and Czech Republic (CZ-Stn). Subsequently, panel (b) offers a magnified view of the outcomes at the CZ-Stn location in 2018, highlighting the extremes defined by a 90% threshold using a greyed-out area.**

[Figure]

(a) Old figure 4          (b) Updated figure 4

**Figure 1:** Side-by-side comparison of Figure 4 after the suggested changes.

6. *Figure5, the legend color in the right panel is missing. Please revise it for completeness.*

The double legend in Figure 5 refers to both panels, with the color legend describing the entropy and complexity measures per 50 runs per model and the black legend describing the mean of the respective models. Your comment made us realize that this is somewhat misleading. We have added a more precise, unified legend to fix this problem. Figure 2 shows the differences between the old and the new Figure 5. We thank the reviewer for pointing out a potential source of confusion.

[Figure]

(a) Old figure 5          (b) Updated figure 5

**Figure 2:** Side-by-side comparison of Figure 5 after the suggested changes.

7. *Table 1. Clarify whether the standard deviation is derived from the 20 selected sites.*

The standard deviation is indeed derived from the study sites. We have revised the legend to state that explicitly. Thank you for pointing this out. We added the following sentence to the caption:

**The means are calculated across 50 runs for each of the 20 different sites, and then the mean of these 20 location-derived means is shown. Similarly, the standard deviations are calculated for each location across 50 runs and then averaged to provide the overall standard deviation for each metric.**

8. *I would suggest adding a comparison of training and inference speed of the selected networks for a more comprehensive evaluation.*

We acknowledge the importance of assessing the performance of neural network architectures in terms of speed. However, due to the models being implemented in different programming languages (Julia for the echo state networks and Python for the backpropagation-based approaches), a direct speed comparison is not representative of differences in the models' performance. In fact, in [3], we show that the Julia implementation of echo state networks offers faster computational times than existing Python packages. Additionally, existing literature (for example [4, 5, 6, 7]) provides extensive comparisons of these architectures' speeds. We have added the following introduction to Appendix A in which we detail existing works comparing these architectures:

**In this appendix, we provide the mathematical details behind the models used in this study. Starting from A1, we introduce the simple version of the RNN, which provides the basic equation for all following models. Subsequently, in A2 and A3, we illustrate the gated approaches of LSTMs and GRUs, respectively. Finally, in A4, we showcase the ESNs. While the general approach for all the models shows some similarities, different constructions exhibit variations in complexity [8, 9] and training speed [4, 5].**

We look forward to hearing from you.
Sincerely and on behalf of all authors,

Francesco Martinuzzi

Leipzig University
Center for Scalable Data Analytics and Artificial Intelligence,
Institute for Earth System Science and Remote Sensing, and
Remote Sensing Centre for Earth System Research

**References**

[1] Basil Kraft, Martin Jung, Marco Körner, Christian Requena Mesa, José Cortés, and Markus Reichstein. Identifying dynamic memory effects on vegetation state using recurrent neural networks. *Frontiers in big Data*, 2:31, 2019.

[2] Richard C. Cornes, Gerard van der Schrier, Else J. M. van den Besselaar, and Philip D. Jones. An ensemble version of the e-obs temperature and precipitation data sets. *Journal of Geophysical Research: Atmospheres*, 123(17):9391–9409, 2018.

[3] Francesco Martinuzzi, Chris Rackauckas, Anas Abdelrehim, Miguel D Mahecha, and Karin Mora. Reservoircomputing. jl: an efficient and modular library for reservoir computing models. *The Journal of Machine Learning Research*, 23(1):13093–13100, 2022.

[4] Luca Cerina, Marco D Santambrogio, Giuseppe Franco, Claudio Gallicchio, and Alessio Micheli. Efficient embedded machine learning applications using echo state networks. In *2020 Design, Automation & Test in Europe Conference & Exhibition (DATE)*, pages 1299–1302. IEEE, 2020.

[5] Chenxi Sun, Moxian Song, Derun Cai, Baofeng Zhang, Shenda Hong, and Hongyan Li. A systematic review of echo state networks from design to application. *IEEE Transactions on Artificial Intelligence*, 5(1):23–37, 2022.

[6] Joschka Herteux, Christoph Räth, Amine Baha, Giulia Martini, and Duccio Piovani. Forecasting trends in food security: a reservoir computing approach. *arXiv preprint arXiv:2312.00626*, 2023.

[7] Xingshang Li, Fanjun Li, Shoujing Zheng, and Qianwen Liu. Nox concentration prediction with a flexible cascaded echo-state network in a cement clinker calcination system. *IEEE Transactions on Industrial Informatics*, 2024.

[8] Pedro J Freire, Yevhenii Osadchuk, Bernhard Spinnler, Antonio Napoli, Wolfgang Schairer, Nelson Costa, Jaroslaw E Prilepsky, and Sergei K Turitsyn. Performance versus complexity study of neural network equalizers in coherent optical systems. *Journal of Lightwave Technology*, 39(19):6085–6096, 2021.

[9] Pedro Freire, Sasipim Srivallapanondh, Bernhard Spinnler, Antonio Napoli, Nelson Costa, Jaroslaw E Prilepsky, and Sergei K Turitsyn. Computational complexity optimization of neural network-based equalizers in digital signal processing: A comprehensive approach. *Journal of Lightwave Technology*, 2024.

---

## Author Comment (AC2)

May 21, 2024

Nonlinear Processes in Geophysics
Manuscript ID: EGUSPHERE-2023-2368

Dear Referee 2,

We thank you for taking the time to read our manuscript and for the helpful comments posted. This letter details the changes we made to address the points raised.

We highlight the referee comments in *blue italics*. We provide our replies to the comments and the changes we made to the manuscript. Additionally, we showcase the manuscript additions and alterations in **bold font**.

1. *I advise the authors to change the places where "temperature" is written to "air temperature". And "global radiation" to "global solar radiation".*

   We thank the reviewer for pointing this out. We have changed the "temperature" to "air temperature" and "radiation" to "global solar radiation" in Lines 28, 86, 136 and 143-144.

2. *Figure 1 a: The scale and north arrow should be added to the map.*

   There is no need to add scale and north arrow since Figure 1a consists of a Robinson-projected map with its corresponding grid, which accounts for representing the varying scale and the non-linear north direction [1].

3. *Description of Fig 1. A typo "radii" needs to be corrected.*

   We appreciate the suggestion, but while both radii and radiuses are viable in English [2], we prefer the former.

4. *L115-116: Did you have in situ NDVI measurements to compare MODIS values?*

   We did not use in situ NDVI measurements since MODIS data provides extensive coverage and captures canopy-level information, which is particularly relevant for our study focused on forested areas. Acquiring in situ measurements across the forested regions would be logistically challenging and resource-intensive (Instruments such as drones equipped with spectroradiometers or multispectral cameras would be required to obtain canopy-level measurements, and ground-based measurements could be influenced by understorey vegetation, leading to discrepancies with the canopy-focused MODIS data). Furthermore, MODIS Nadir BRDF Adjusted Reflectance (NBAR) data has been validated and is a reliable source for large-scale vegetation analysis [3].

5. *Figure 2: It would be good to show differences in network architectures.*

   We appreciate the request to add more clarity in one key figure of the methods section. At the same time, we want to maintain the focus on the research question, which focuses on learning vegetation dynamics. We purposely kept the figure simple to give a quick intuition of the main differences in the models, i.e., the different training concepts. More details are shown in the caption and in the text itself in Sections 2.3 and 2.4. We outline mathematical details in appendices A1 through A4, providing different levels of detail for the interested readers. We have added titles to the figures, clarifying the differences between the backpropagation approach to training and the echo state approach. Figure 1 shows the differences between the old and updated versions of Figure 2. In addition, we have now included references in the introduction of Appendix A, which solely focuses on architecture comparison, for which toy models and benchmark datasets are standard.

[Figure]

(a) Old figure 2          (b) Updated figure 2

**Figure 1:** Side-by-side comparison of Figure 2 with updated subtitles.

6. *Section 2.4: What is the ratio between training and testing data?*

   Thank you for addressing this. While we mention the training and testing data length in the text, it is only done so once and not in the proper section. Additionally, we do not mention the training and testing ratio. To address this problem, we added the following sentences at the end of Section 2.1:

   **We split the data for training and testing as follows: years from 2000 to 2013 (inclusive) for training and validation. We use the remaining years, from 2014 to 2020, for testing, resulting in a 67% and 33% training and testing split ratio.**

7. *Figure 3e: Please explain the meaning of the red-shaded area here?*

   We added the following line to the figure's caption to avoid confusion.

   **The extreme responses are highlighted by a red shaded area.**

8. *Section 2.6.1: Please add a statement indicating at which values these performance indicators work well.*

   The metrics indicate improved model performance as they approach zero. However, their true significance is best understood by comparing them with the same metrics from other models. We added a few sentences at the end of subsection 2.6.1, Line 248, to emphasize this context.

   **Both metrics proposed in this section indicate better model performance as their value approaches zero.**

9. *L246: Please use another abbreviation for entropy-complexity. In Line 117, EC stands for "eddy covariance".*

   Thanks for pointing out this oversight. We have now removed the use of acronyms for eddy covariance towers.

10. *Figure 5b: The legend should be corrected.*

    We have corrected the legend of Figure 5. Figure 2 shows the differences between the old and new legends.

11. *Appendix B: Please add computational resources (GPU, etc.)?*

    We thank the reviewer for this query about computational resources used in the paper's experiments. To clarify the setup, we included the following information in Appendix B1:

    **All the simulations are run on a machine fitted with an NVIDIA RTX A6000 graphics processing unit (GPU) and 504 GB of random access memory (RAM).**

12. *Data Availability section: You should add MODIS data source link.*

    Our study exclusively used the curated dataset FluxNetEO, which is based on MODIS data, see section 2.1 Lines 115. We believe we provided all the necessary links and citations for

[Figure]

**(a) Old figure 5**        **(b) Updated figure 5**

**Figure 2:** Side-by-side comparison of Figure 5 after the suggested changes.

this study to be reproducible.

We look forward to hearing from you.
Sincerely and on behalf of all authors,

Francesco Martinuzzi

Leipzig University
Center for Scalable Data Analytics and Artificial Intelligence,
Institute for Earth System Science and Remote Sensing, and
Remote Sensing Centre for Earth System Research

**References**

[1] John P. Snyder. *Map projections: A working manual*. 1987.

[2] Radius definition and meaning. https://www.merriam-webster.com/dictionary/radius. Accessed: 2024-05-13.

[3] Crystal Schaaf and Zhuosen Wang. Modis/terra+aqua brdf/albedo nadir brdf adjusted ref daily l3 global - 500m v061. 2021.

---

## Author Response (AR2)

August 1, 2024

Nonlinear Processes in Geophysics
Manuscript ID: EGUSPHERE-2023-2368

Dear Dr. Zoltan Toth, dear Reviewer 2,

We thank the editor, Dr. Zoltan Toth, for reading through our paper, the replies, and the reviews and for the additional feedback provided. Additionally, we thank Reviewer 2 for taking the time to address our replies to previous reviews and for providing additional comments. This letter details the changes made to the manuscript to address the points raised.

We highlight the referee comments in *blue italics*. We provide our replies to the comments and the changes we made to the manuscript in **bold font**. In the manuscript, changes are highlighted in red.

1. *Title. Does the "extremes" refer to climate drivers or vegetation response? I assume it should be vegetation responses if my understanding is correct.*

   That is correct, extremes in the title refers to the vegetation response.

2. *Neural network training. I am concerned about the training method and do not agree with the authors' claim about the "universal model". See these relevant studies: [1] and [2] It has been proven that a diverse dataset would increase the model performance.*

   We thank the reviewer for providing us with interesting and relevant literature. Further reading and discussion into the topic helped us realize that, although 20 locations are fewer compared to other studies, their impact on the overall performance should still be noticeable, as noted in [1], Figure 5. Therefore, we believe that a global model with additional features would demonstrate increased performance, as shown in previous research [3], even when trained with just the 20 locations used in our study.

   Our goal was to investigate the immediate response of vegetation to climate data on a site level with a focus on extremes. We choose to train on specific locations in order to highlight the difference of the models in simple settings, which can be increased in complexity in follow up studies. We now see that this point is not stressed in the manuscript.

   Lastly, similar work to the one kindly provided by the reviewer is not yet available in the vegetation community. It would be interesting to investigate the differences using a global approach versus a local one for general accuracy and prediction of extremes in vegetation responses, especially given the importance that memory effects play in biosphere dynamics [4] and the effects extremes have on its resilience [5]. We agree with the editor in the assessment that our study represents a first step in this direction. To this end, we have added a paragraph in the discussion section (Lines 420-431):

   **Most existing studies center on creating a single global model, which is key to understanding performances beyond the training sample [3, 6]. However, we opted to train individual models for each site to understand how, under optimal conditions, each architecture could capture the specific dynamics of an ecosystem without the need for extrapolation. In this way, we could focus on the models' inherent capability to understand each ecosystem's dynamics, and we are sure that model differences do not arise due to different data demands for generalizations. Increasing the locations, features and expanding the models are necessary future steps for the continued investigation of modeling extremes in vegetation with ML. It has been shown that including additional features and multiple locations improves the ML model performance in hydrological applications [1, 2]. While similar studies do exist in the context of biosphere dynamics [3], more investigation is needed. For example,**

**including more locations, as highlighted in [1] represents the easiest way to improve performance. Investigating the effects of more location on the performance of ML for vegetation extremes would be an important contribution for the continued adoption of ML models in predicting biosphere dynamics.**

3. *In addition, for the current training setup. How did the authors tune the hyperparameters? Do all the sites share the same set of hyperparameters? Line 139-141 mentioned 2000 to 2013 is used for training and 2014 to 2020 for testing, suggesting there is no standalone validation dataset for hyperparameter tuning. Section B2 mentioned "Split temporal cross-validation". How many folds are used here? Please clarify the hyperparameter tuning method.*

Hyperparameters are selected through temporal cross-validation for each site. Sites do not share the same set of hyperparameters. For the temporal cross-validation, we used three folds, and for each fold, 20% of the training data was reserved for validation. We appreciate the comment pointing out that our explanation in the text lacked details. We added the following text to provide more context for the training of our models (Lines 501-502):

**We used three folds for cross-validation, with 20% of the training dataset left for validation in each fold.**

4. *In Section B1, 50 out of 100 runs are selected based on performance. Do these 100 runs share the same set of hyperparameters? If so, is this process a selection of initial weights? An ensemble prediction is reasonable but cannot be used for models/runs selections since the testing set should be treated unseen.*

We thank the reviewer for this observation. The 100 runs share the same set of hyperparameters obtained with the tuning selection described above. The different runs are initialized with different weights. We now include the results of all 100 runs and have updated Table 1, Figures 5, 6, 7, A1, and section B1. This does not change the results qualitatively, as can be seen from the updated table and figures.

5. *Input features. Line 153 mentioned T is the time window after which only the input variables are available. What's the value of T in this study? Is it only concurrent variables as input (e.g., temperature at day 10 to predict NVDI at day 10), or T is specified to construct a time window (e.g., temperature from day1 to day T to predict NVDI at day T)? Is it a sequence-to-sequence model or sequence-to-one model? Please clarify it.*

It only uses concurrent variables as inputs, and T represents the length of the training dataset. We claryfied some notation in the technical description of the setup. We thank the reviewer this this observation.

6. *Line 155. What's the purpose of introducing "observer"? It seems irrelevant.*

Thanks for poiting this out, we removed the observer explanation in the revised manuscript.

7. *Metrics for extremes. I would suggest considering F1-score (or area under ROC curve) since these metrics consider true positives, false negatives, and false positives at the same time.*

Thanks for pointing out alternative metrics to use for studying the extremes. We added the F1 scores for the single locations in the appendices. This further showcases the different performances of the models for each site.

8. *Figure 6, 7. Are these figures for all the sites? Is there any performance difference among the sites?*

These figures include results from all the sites. This is now clarified in the figure captions. Additionally, we now provide a figure in Appendix C (Fig B1), which shows the F1 scores for all the unique locations for all different quantiles studied. Figure 1 shows the added figure to Appendix C.

We look forward to hearing from you.
Sincerely and on behalf of all authors,

Francesco Martinuzzi

Leipzig University
Center for Scalable Data Analytics and Artificial Intelligence,
Institute for Earth System Science and Remote Sensing, and
Remote Sensing Centre for Earth System Research

**References**

[1] F. Kratzert, M. Gauch, D. Klotz, and G. Nearing. Hess opinions: Never train an lstm on a single basin. *Hydrology and Earth System Sciences Discussions*, 2024:1–19, 2024.

[2] Kuai Fang, Daniel Kifer, Kathryn Lawson, Dapeng Feng, and Chaopeng Shen. The data synergy effects of time-series deep learning models in hydrology. *Water Resources Research*, 58(4):e2021WR029583, 2022.

[3] Basil Kraft, Martin Jung, Marco Körner, Christian Requena Mesa, José Cortés, and Markus Reichstein. Identifying dynamic memory effects on vegetation state using recurrent neural networks. *Frontiers in big Data*, 2:31, 2019.

[4] Kiona Ogle, Jarrett J Barber, Greg A Barron-Gafford, Lisa Patrick Bentley, Jessica M Young, Travis E Huxman, Michael E Loik, and David T Tissue. Quantifying ecological memory in plant and ecosystem processes. *Ecology letters*, 18(3):221–235, 2015.

[5] Jaboury Ghazoul, Zuzana Burivalova, John Garcia-Ulloa, and Lisa A King. Conceptualizing forest degradation. *Trends in ecology & evolution*, 30(10):622–632, 2015.

[6] Codruț-Andrei Diaconu, Sudipan Saha, Stephan Günnemann, and Xiao Xiang Zhu. Understanding the role of weather data for earth surface forecasting using a convlstm-based model. In *Proceedings of the IEEE/CVF Conference on Computer Vision and Pattern Recognition*, pages 1362–1371, 2022.

[Figure]

Figure 1: Additional figure in Appendix C.

---

## Author Response (AR3)

September 6, 2024

Nonlinear Processes in Geophysics
Manuscript ID: EGUSPHERE-2023-2368

Dear Dr. Zoltan Toth, dear Reviewer,

We thank the editor, Dr. Zoltan Toth, for the helpful comments in the revised version of the paper. We additionally thank the reviewer for reading and evaluating our manuscript.

We highlight the editor comments in *blue italics*. We highlight the changes and additions we made to the manuscript in **bold font**.

1. *In your response to Rev. 2's question about the current title "Learning Vegetation Response to Climate Drivers and Extremes with Recurrent Neural Networks" you indicate that "extreme" refers to "vegetation response". The current title does not reflect that. Would "Identifying Extreme Vegetation Response to Climate Drivers with Recurrent Neural Networks", or something similar be more expressive? This may be a shorter, more to the point title?*

   We agree that the title could be misunderstood, and we appreciate the suggestion for its modification. We feel that changing it to **Learning Extreme Vegetation Response to Climate Drivers with Recurrent Neural Networks** better reflects the contents of the manuscript, while also being unambiguous.

2. *In response to Rev. 2's 3rd comment, you introduced the term "fold" in Appendix B2. This word is not defined and sounds like jargon. Can you please define "fold" before its first use, or possibly replace with a more commonly used / self-explanatory expression?*

   Thank you for pointing out the use of *fold* without a proper introduction to it. We have added some more explanation to the temporal cross-validation technique used and the meaning of the terminology in Lines 508-510:

   **In this technique, the training data is subdivided into smaller sections of increasing length, called folds. Each fold contains both training and testing data, with a chosen split between them.**

We look forward to hearing from you.
Sincerely and on behalf of all authors,

Francesco Martinuzzi

Leipzig University
Center for Scalable Data Analytics and Artificial Intelligence,
Institute for Earth System Science and Remote Sensing, and
Remote Sensing Centre for Earth System Research